# Ovarian Fibrothecoma in a Mare—Case Report

**DOI:** 10.3390/ani14091307

**Published:** 2024-04-26

**Authors:** Raimonda Tamulionytė-Skėrė, Nomeda Juodžiukynienė, Renata Gruodytė, Paulina Rimkutė, Iveta Duliebaitė, Akvilė Savickytė

**Affiliations:** 1Large Animal Clinic, Lithuanian University of Health Sciences, 44307 Kaunas, Lithuania; renata.gruodyte@lsmuni.lt (R.G.); paulina.rimkute@lsmuni.lt (P.R.); iveta.duliebaite@lsmuni.lt (I.D.); akvile.savickyte@lsmuni.lt (A.S.); 2Department of Veterinary Pathology, Lithuanian University of Health Sciences, 44307 Kaunas, Lithuania; nomeda.juodziukyniene@lsmuni.lt

**Keywords:** fibrothecoma, ovarian tumor, neoplasia, histopathology

## Abstract

**Simple Summary:**

A 6-year-old Dutch Warmblood (KWPN) mare was taken to the Lithuanian University of Health Sciences Large Animal Clinic with colic signs. During a diagnostic laparotomy, an altered right ovary was found and removed. Macroscopically, one part of the ovary was hard and gray. The other parts were smaller, soft, and yellowish, with abundant hemorrhages. Ovarian neoplasia was suspected. During the microscopical examination, the tumor showed a mixed histological appearance. The predominant population of cells showed ovoid to round nuclei and a pale-gray, abundant, oval, round cytoplasm, and indistinct cell membranes imparted a syncytial appearance. A diffuse growth pattern was observed. The cell nucleus showed minimal atypia with large, distinct nucleoli and a moderate mitotic index (10 mitosis). The mentioned features were consistent with fibrothecoma. The tumor was mixed in part of the thecoma in the region of the borders or showed a sharp border in places. The fibroma showed a very fine storiform pattern, which was composed of uniform spindle-shaped cells with an oval nucleus and minimal atypia. The normal histological structure of the ovary was fully destroyed—no primordial, primary, secondary, or tertiary follicles were found. The usual ovarian stroma also was not detected. Based on the histological findings, the tumor was consistent with ovarian fibrothecoma.

**Abstract:**

Ovarian tumors in mares are uncommon in comparison to other neoplasms and are classified into three categories: gonadal stromal tumors, coelomic epithelium surface tumors, and germinal cell tumors. Some ovarian neoplasms histologically show a mixture of multiple cell types in the same tumor, such as fibrothecoma; therefore, the differentiation between fibroma and thecoma is often difficult. According to the World Health Organization, fibrothecomas are classified as sex-cord stromal tumors (pure stromal tumors). Neoplasms such as fibrothecoma present with limited morphological, clinical, ultrasonographic, and endocrine profile characteristics. To diagnose this type of tumor, a broad clinical examination is needed, but histopathology remains the most accurate. Herein, we report a case of incidentally found ovarian fibrothecoma during a diagnostic laparotomy in a 6-year-old Dutch Warmblood (KWPN) mare who presented to the clinic with colic symptoms. After a unilateral ovariectomy, the altered right ovary was diagnosed as fibrothecoma based on histopathological features.

## 1. Introduction

Fibrothecoma is the rarest ovarian tumor type for equids and is seldom reported in mares. There is limited information about equine fibrothecoma in the veterinary literature. In human medicine, fibrothecomas represent only 1–4.7% of ovarian tumors [1,2]. Clinical signs of ovarian neoplasms can be nonspecific [3]. In human medicine, these tumors are asymptomatic, usually detected during routine gynecologic examination [4,5]. Fibrothecoma is a benign neoplasm that can be hormonally active and can secrete estrogen and androgens [3]. Estrogen production can cause infertility by inducing defects in ovulation or implantation [6]. However, hormone assays can show a normal concentration of testosterone and progesterone [7]. According to the literature, apart from behavior and estrous changes, fibrothecoma may present as infertility, weight loss, or colic [8]. There is no information about its malignancy in the veterinary literature [4]. This study aims to report a case of ovarian fibrothecoma in a mare.

## 2. Case History

In December 2023, a 6-year-old KWPN mare presented with colic symptoms (apathy, depression, refusing to eat and drink, no passing of manure) at the Lithuanian University of Health Sciences Large Animal Clinic. According to the history, the mare experienced colic a month ago, which was resolved solely through a conservative approach. The mare had a low body score (2 out of 5). Upon physical examination, the heart and breathing rate, rectal temperature, mucous membranes, and capillary refill time were within normal ranges, except for weakened peristalsis in the small colon. No significant changes were observed during the abdominal ultrasound. Hematological examination revealed lymphopenia (1.48–10^9^/L, RI 1.5–7.7), leukopenia (4.45–10^9^/L, RI 5.40–14.30), hypochromia (MCHC 296 g/L, RI 310–390), and thrombocytopenia (99–10^9^/L, RI 100–400). Serum biochemistry was within the physiological limits. Blood lactates were 1.8 mmol/L (RI 1–1.5) (LACTATE PRO 2, Arkray Inc., Kyoto, Japan), glucose was 4.9 mmol/L (RI 4.3–5.5) (eBGK-VET, VISGENEER Inc., Hsinchu, Taiwan), and PCV was 37%. Perirectal palpation revealed formed, very hard, and dry feces in the ampulla and small colon; also, the edge of the spleen was slightly displaced medially from the abdominal wall. The rectal palpation of the uterus and ovaries did not reveal any significant changes. The mare did not show any discomfort during the rectal palpation of the reproductive system. Five liters of water and electrolytes were provided through a nasogastric tube to the stomach. Intravenous fluid therapy (Ringer lactate, B. Braun, Melsungen, Germany) and metamizole natrium salt (Biowet Drwalew, Drwalew, Poland; 30 mg/kg bw i.v.) were supplied.

On consecutive days, intravenous fluids (an average of 10–15 L every 12 h) and pain relief therapy were administered, along with fluids through the nasogastric tube, as the mare refused to drink. Blood lactates decreased to 1.3, 1.1, and 0.9 mmol/L; however, the clinical condition did not change significantly. Motility of the small colon and small intestines decreased; nevertheless, the mare was fed a small amount of hay. After a few days, the mare started to show acute colic symptoms (restlessness, pawing the ground, no passing of manure, apathy). An elevated heart rate (55 beats per minute) was observed. Ultrasonographically, 1 cm of anechoic free fluid within the peritoneal cavity was detected; a slightly dilated duodenum and decreased motility of the small colon loops were observed at the lower flank bilaterally.

Abdominocentesis was performed. The peritoneal tap was opaque and dark-yellow (Figure 1). Peritoneal tap glucose was 11.3 mmol/L (RI 4–7) (eBGK-VET, VISGENEER Inc., Taiwan), lactates 1.6 mmol/L (RI 0.4–1.2) (LACTATE PRO 2, Arkray Inc., Japan), and total proteins 4 g/dL. Based on exacerbated clinical symptoms, the free fluid present within the peritoneal cavity, and the abdominocentesis results (increased glucose, lactates, and total protein levels and changes in the peritoneal tap color and turbidity), it was decided to perform a diagnostic laparotomy.

The mare was premedicated with xylazine 20 mg/mL (Biowet Drwalew, Poland; 1.1 mg/kg bw i.v.) and positioned in dorsal recumbency after induction with ketamine 10% (VetViva Richter GmbH, Wels, Austria; 2.2 mg kg/bw) and midazolam (Kalceks, Riga, Latvia; 0.05 mg kg/bw) and was maintained with sevoflurane (Chanelle Pharmaceuticals Manufacturing Ltd., Galway, Ireland) delivered in 100% oxygen. The surface of the abdomen was prepared aseptically, and a ventral midline celiotomy was performed. An incision of approximately 30 cm was made. During the routine exploratory laparotomy, no intestinal dislocation was present. Enlargement of the spleen was observed. During the diagnostic laparotomy, an altered right ovary was found incidentally, and neoplasia was suspected. The left contralateral ovary had a normal appearance and size. After right ovary exposure, transfixion ligations of the ovarian pedicle were made with PGA USP 3&4 (SMI, Vith, Belgium), the ovarian pedicle was sectioned, and the ovary was removed from the peritoneal cavity. Finally, a routine three-layer closure of the abdominal wall was performed.

The excised mass underwent histopathologic examination. Tissue samples were fixed with 10% formalin solution and taken to the Pathology Center of the Veterinary Academy of the Lithuanian University of Health Sciences.

After surgery, the mare was treated with benzylpenicillin (Sandoz, Basel, Austria; 33,000 U.I. kg/bw i.v. BID) and gentamicin (Dopharma, Raamsdonksveer, The Netherlands; 6.6 mg kg/bw i.v. SID) for five days. Flunixin meglumine (Vet-agro, Lublin, Poland; 1.1 mg kg/bw i.v.) was administered for postoperative pain management. Due to a postoperative lack of peristalsis, neostigmine (Cenavisa S.L., Reus, Spain; 0.0044 mg kg/bw s.c.) was administered three hours post-surgery and once on the second day after surgery. Fluids were applied through the nasogastric tube 3–4 times a day for the first two days, due to refusing to drink postoperatively.

Topical treatment of the incision was performed daily using povidone–iodine (Valentis, Vilnius, Lithuania). No complications were observed, and sutures were removed within 14 days.

After surgery, the mare’s reproductive system was investigated. Ultrasonographically, slight uterine edema (grade 1 out of 5) and hyperechoic particles (Figure 2), suspected as air in the uterus, were observed. The left ovary was normal in size and echogenicity, present with some 10 to 28.4 mm follicles (Figure 3 and Figure 4). Due to poor perineal conformation, the mare had pneumovagina. Caslic’s vulvoplasty was performed.

## 3. Diagnosis

On gross examination, the removed ovary was approximately 10 × 4 × 5 cm. Macroscopically, the right ovary was enlarged as compared to the contralateral ovary. The whole ovary was effaced and replaced entirely by the neoplasm. A major part of the tumor was hard and gray (Figure 5). The other parts of the tumor were smaller, soft, and yellowish, with abundant hemorrhages (Figure 6). Both macroscopically and histologically, two clearly distinct tumor zones were visible.

A histological examination was performed. Paraffin blocks were made using Shandon Pathcentre (Thermofisher, Waltham, MA, USA) and TES99 (Medite Medizintechnik Burgdorf, Germany) equipment. Four-micrometer sections were obtained using the Shandon Pathcentre (Thermofisher, Waltham, MA, USA) microtome. Serial 4 μm sections were prepared with a Sakura Accu-Cut SRM microtome (Sakura Finetek, Torrance, CA, USA).

Sections were stained with the routine hematoxylin–eosin (HE) technique using Sakura Accu-Cut SRM (Japan) equipment (Sakura Finetek, Torrance, CA, USA). Histological slides were evaluated using the Olympus microscope supplied with a digital Olympus DP72 image camera with CellSensDimension software, version 1.14.

Histologically, both tumor areas were distinctly different from each other (Figure 7 and Figure 8). In the obtained sample for histological examination, the portion of the thecoma was significantly lower (Figure 9). Marked hemorrhage (but not blood-filled lacunae) and very sparse granulation-type connective tissue were observed at the border between both tumor tissues (Figure 10). Hemorrhagic blood masses infiltrated a larger part of the thecoma tissues. A single cystic structure lined with columnar epithelium was found in the hemorrhagic zone, with a very sparse amount of mucus-like material in the lumen (Figure 11).

Thecoma cells had changed the nucleus/cytoplasm ratio in favor of cytoplasm (cytoplasm of thecoma cells was very abundant). The cytoplasm of thecoma cells was irregular in shape and amount, with distinct borders and different sizes and amounts of vacuoles. Nuclei were large, with moderate size and shape variation and large, irregular size nucleoli (macronucleoli) (Figure 12). The mitotic rate was moderate in thecoma tissues—10 normal mitoses per 10 fields at 400 magnification (Figure 13).

The fibroma was composed of uniform medium-sized, spindle-shaped cells with indistinct borders and a regularly shaped and sized oval, dark or gray nucleus to long thin nucleus cells. The indistinct nucleoli and finely stippled chromatin in the nucleus were observed. The nucleus/cytoplasm ratio favored the cytoplasm. The cytoplasm was pink and long, as is common in mature spindle cells. Spindle cell anisocytosis and anisokaryosis were mild to moderate (Figure 14). Mitoses were typical. The rate of mitoses was low—2 per 10 fields at 400 magnification. The spindle-like cells were arranged in sweeping fascicles angled in a chevron-like or herringbone pattern (Figure 15). In addition, storiform areas were observed (Figure 16). The stroma had a delicate intercellular network of collagen, but a few areas of “keloid-like” sclerosis/hyalinization were found (Figure 17). The histological pattern is typical for fibrosarcoma, but not for fibroma. However, tumor cells lacked marked atypia. The ovary-derived fibroma was considered to be a pure stromal tumor of the ovary.

In both specimens, the conventional ovary tissue area was absent. The normal histological structure of the ovary was fully destroyed—no primordial, primary, secondary, or tertiary follicles were found. In addition, the usual ovary stroma was not detected.

## 4. Discussion

Macroscopically, fibrothecoma is a variably sized, whitish or yellowish [9]. neoplasia with a thick wall and a solid and firm consistency. The fibrothecoma surface varies from regular [9] to irregular [10]. Areas of hemorrhage and necrosis are also evident [11]. Hemorrhagic areas were observed both inside [11] and outside of the neoplasia.

Histologically, fibrothecoma consists of oval- or spindle-shaped cells and intersecting fascicles [9]. Lipid vacuoles and the presence of hemosiderin can also occur [10]. Vesicular nuclei vary from ovoid to fusiform in shape; they are slightly eosinophilic and foamy, vacuolated cytoplasms [10] with poorly defined borders. Mitotic figures were not evident [3]; in other cases, the mitotic amount was less than 2 cells per 10 HPF [9]. Well-circumscribed and variably sized vascular spaces filled with erythrocytes are also characteristic [9,12].

However, both macroscopic and microscopic views of fibrothecoma are very variable. Histopathologically, some ovarian neoplasms show a mixture of multiple cell types in the same tumor [4,10]. Sometimes it is impossible to differentiate between thecoma and fibroma, so the term “fibrothecoma” can be used to describe neoplasms with altered theca cells and fibroma characteristics. Both the macroscopic and microscopic views corroborated this case report.

To diagnose fibrothecoma, more specific tests are required: histopathology, immunohistochemistry, and hormonal-level testing [3]. Due to the relatively nonspecific features, a definitive diagnosis is mostly based on histological findings [5]. In the present case, due to the incidental finding during an exploratory laparoscopy, it was decided that only a histopathological examination would be performed.

The clinical signs of ovarian tumors are relatively nonspecific. In horses, apart from reproductive behavior changes, masculinization, persistent anestrus [5], and anovulatory follicle persistence during the breeding season, fibrothecoma often appears clinically asymptomatic [6]. It can also occur as progressive weight loss [12] anorexia, and pyrexia [12]. It has been reported that ovarian neoplasms in a mare may cause colic [8,13]. In this case, we hypothesized that ovarian neoplasm fibrothecoma in the right ovary may be a reason for the low body score and possibly cause colic symptoms for this mare.

Fibrothecomas may or may not interfere with steroid production and levels [14]. They might be endocrinologically functional and produce testosterone and inhibins. Fibrothecoma might produce estrogen and cause infertility by inducing defects in ovulation or implantation [1,2]. Serum progesterone, estradiol, testosterone, and Anti-Müllerian hormone assays can be useful adjuncts for detecting hormone-producing ovarian neoplasms; however, some authors have reported hormone levels within physiological limits [11]. Here, fibrothecoma was an incidental finding as the mare did not show any obvious changes in behavior such as masculinization or changes in the estrous cycle, so no hormonal assays were performed in this case.

Hormone-producing neoplasms in mares can influence reproductive behavior, which may range from estrogenic activity to masculinization [12]. The most common signs of hormonally productive sex-cord tumors are stallion-like behavior and infertility [15]. In this case, the owners observed only excessive vocalization to other horses. Additionally, the mare showed no masculine physical changes such as a large clitoris or a heavily crested neck [3]. No other stallion-like behavior was observed, as the owners did not use the mare for reproductive purposes.

In animals, ovarian neoplasms are more prominent in elderly rather than in young horses [10,14]. However, there are reported cases where thecoma and fibrothecoma occurred in young, 4-to-5-year-old mares [3,16]. In this clinical case, fibrothecoma was diagnosed in a young, 6-year-old mare.

Metastasis in sex-cord neoplasms is relatively common and is reported in bitches, queens, and cattle [7,17]. Fibrothecomas with malignancy are rare [14]. Thus, metastases can occur in regional lymph nodes via the bloodstream and can implant in the peritoneal cavity [14,17] even though, in many cases, such as that of Raoofi A. et al. [3] no signs of metastasis were detected. In the present case, no metastatic signs were detected macroscopically, either.

In mares, information about reproductive life after fibrothecoma removal is lacking. The prognosis for ovariectomy is considerably successful, with the mare regaining reproductive health a few months post-unilateral ovariectomy [18] Some mares may become infertile, with a small, contralateral ovary [6,14]. However, there is a reported case in which the mare returned to the estrous cycle after unilateral ovariectomy due to fibrothecoma and was successfully inseminated with a positive embryo transfer [6]. As far as the authors are aware, the mare had one foal earlier. In the present case, further investigation of the reproductive system is needed to evaluate the mare’s fertility after unilateral ovariectomy.

There are several potential limitations in the management of this clinical case: no ultrasound investigation of the reproductive system was performed because the mare did not have a history of estrous disorders and did not show any discomfort during rectal palpation; thus, an ultrasonographic examination was not indicated in this case. No endocrinological investigations were conducted because the neoplasm was observed incidentally during a diagnostic laparoscopy.

## 5. Conclusions

In this study, the diagnosis of fibrothecoma was based on histopathologic observations. Sex-cord stromal tumors such as fibrothecomas have seldom been reported in mares [10], so further investigations are needed to increase our knowledge of the recognition, diagnosis, and prognosis of reproductive potential post-unilateral ovariectomy.

## Figures and Tables

**Figure 1 animals-14-01307-f001:**
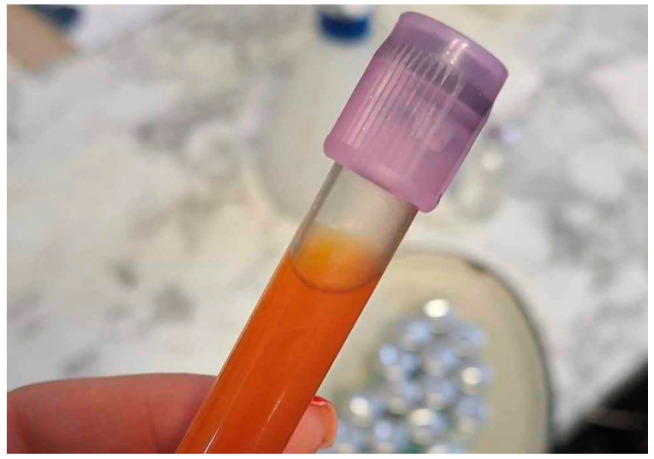
Dark yellow and opaque peritoneal tap.

**Figure 2 animals-14-01307-f002:**
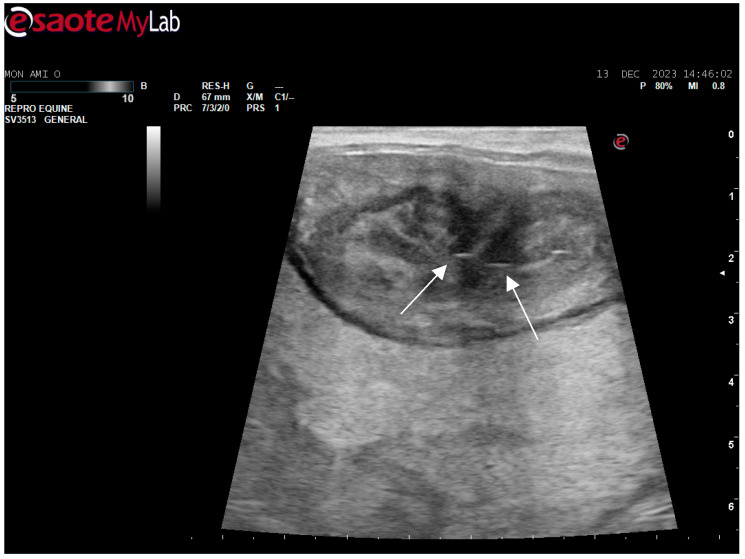
Uterine edema (grade 1 out of 5) and hyperechoic particles (white arrows), suspected as air in the uterus.

**Figure 3 animals-14-01307-f003:**
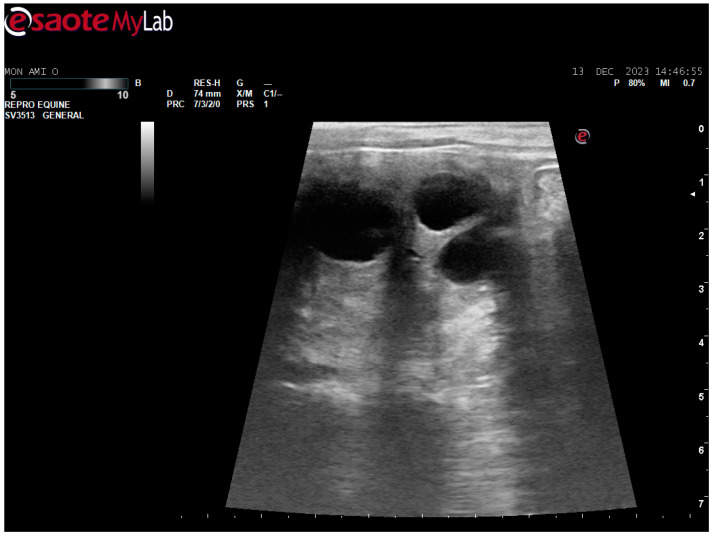
Left ovary, present with some >10 mm follicles. Ultrasonographic picture post surgery.

**Figure 4 animals-14-01307-f004:**
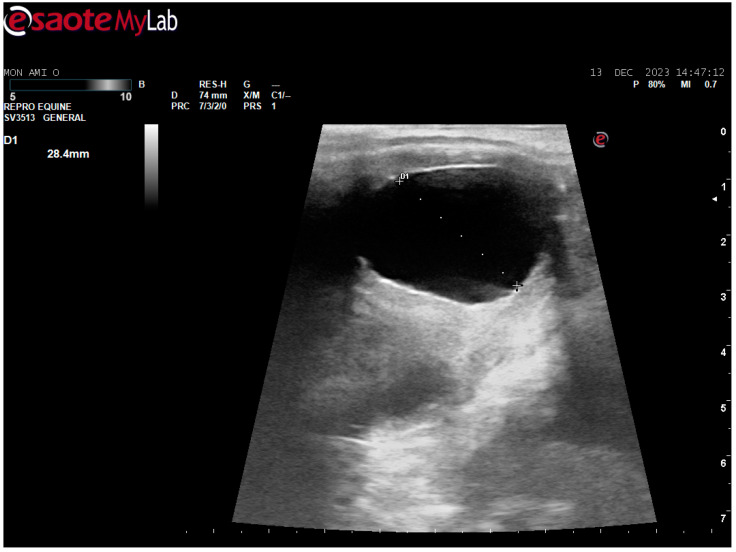
Left ovary, present with 28.4 mm follicle follicles. Ultrasonographic picture post surgery.

**Figure 5 animals-14-01307-f005:**
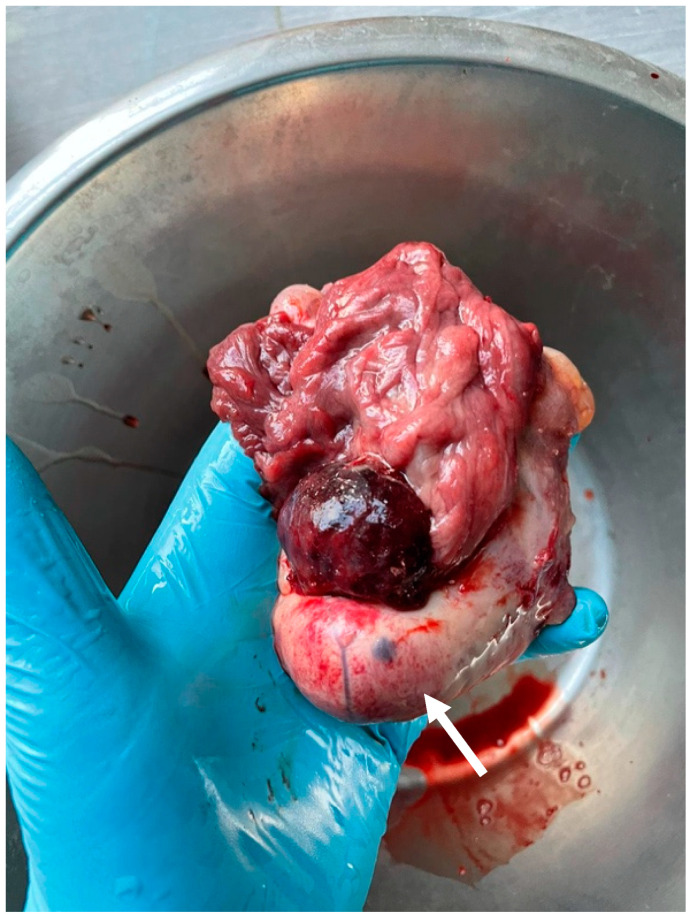
Macro view of mare’s right ovary. Arrow shows very hard, gray ovarian edge.

**Figure 6 animals-14-01307-f006:**
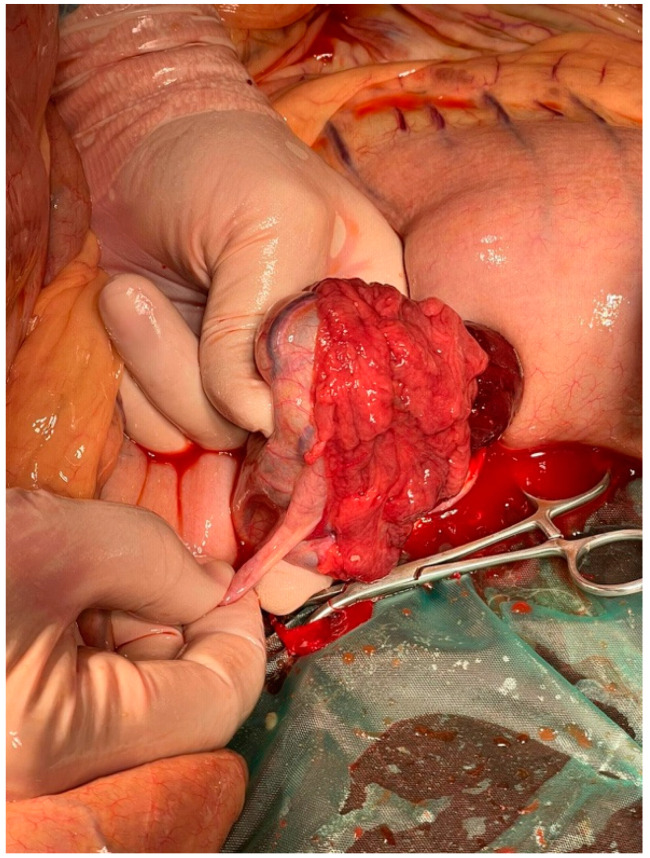
Macro view of mare’s right ovary during diagnostic laparotomy.

**Figure 7 animals-14-01307-f007:**
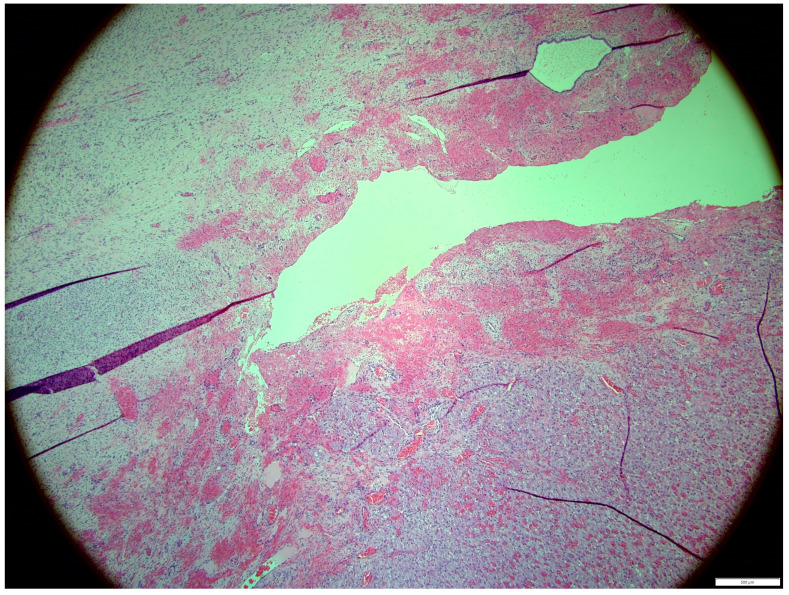
Distinction between fibroma and thecoma parts. HE, 100× magnification.

**Figure 8 animals-14-01307-f008:**
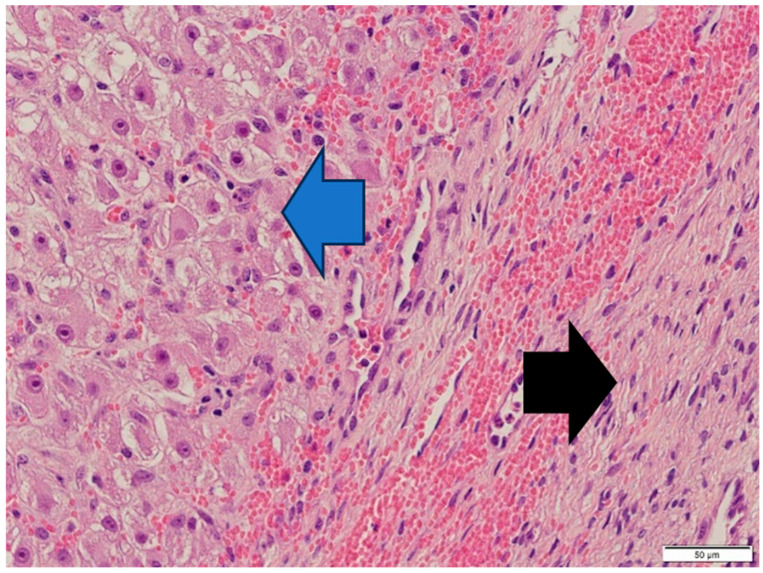
Thecoma: population of cells showing ovoid to round nuclei and pale gray abundant oval, round cytoplasm. The distinct large nucleolus inside the nucleus is present (blue arrow). HE, 200× magnification. Black arrow—part of fibroma.

**Figure 9 animals-14-01307-f009:**
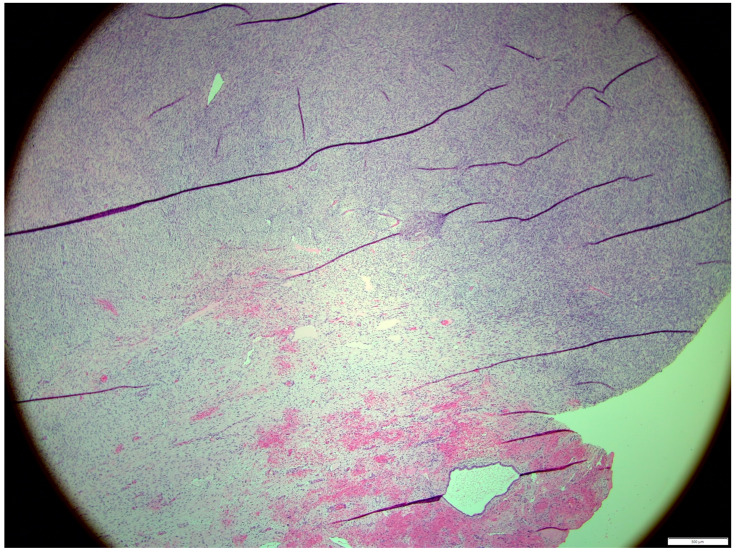
The portion of the thecoma was significantly lower. HE, 100× magnification.

**Figure 10 animals-14-01307-f010:**
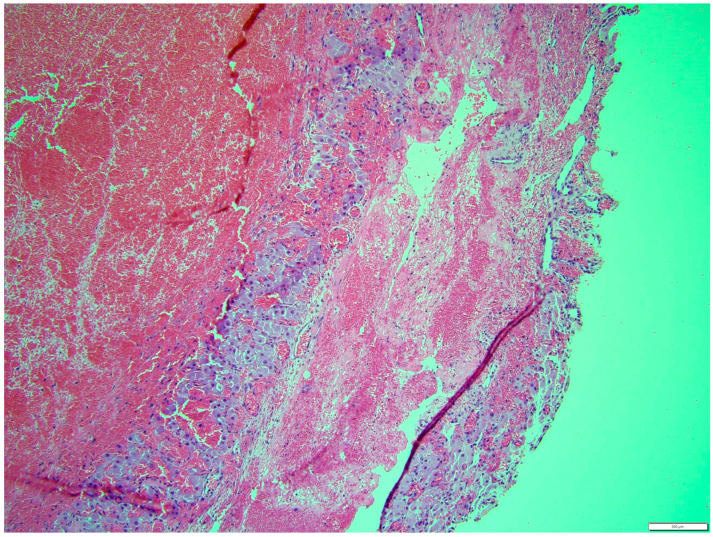
Marked hemorrhage (but not blood-filled lacune) and very sparse granulation-type connective tissue were observed at the border between both tumor tissues. HE, 200× magnification.

**Figure 11 animals-14-01307-f011:**
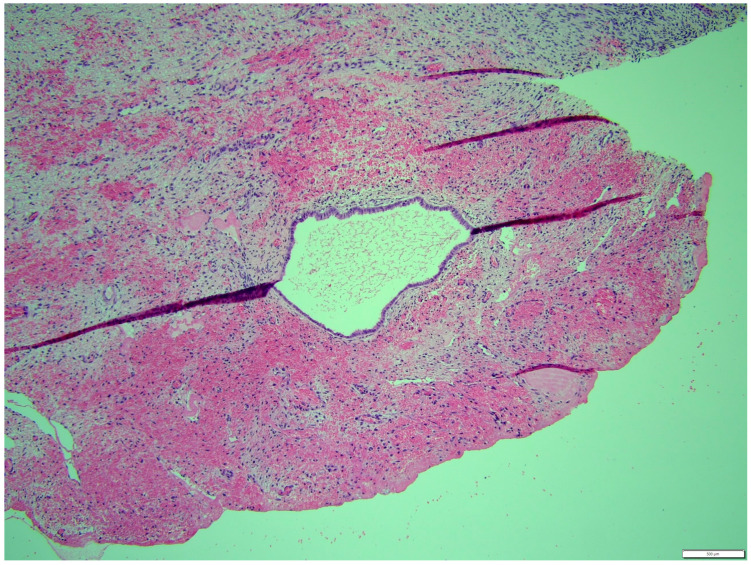
A single cystic structure lined with columnar epithelium was found in the hemorrhagic zone, with a very sparse amount of mucus-like material in the lumen (thecoma part). HE, 100× magnification.

**Figure 12 animals-14-01307-f012:**
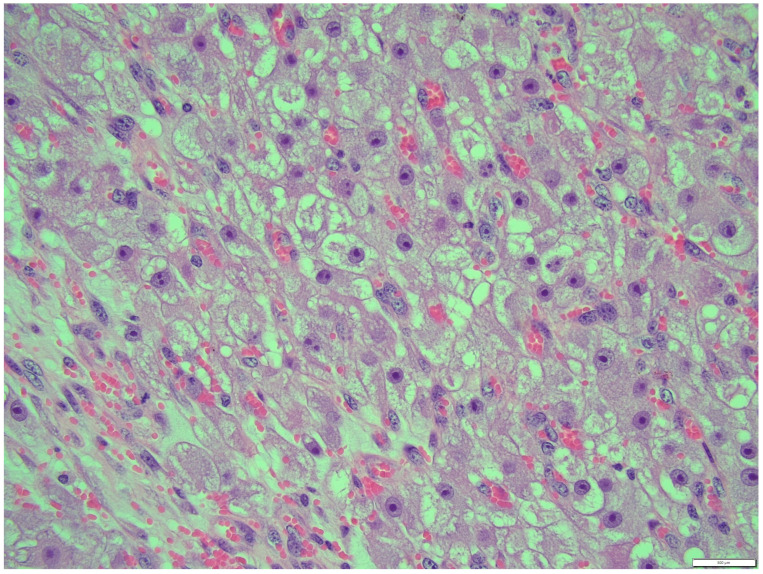
Nucleus were large, with moderate size and shape variation, with large, irregular size nucleolus (macronucleoli), thecoma part. HE, 400× magnification.

**Figure 13 animals-14-01307-f013:**
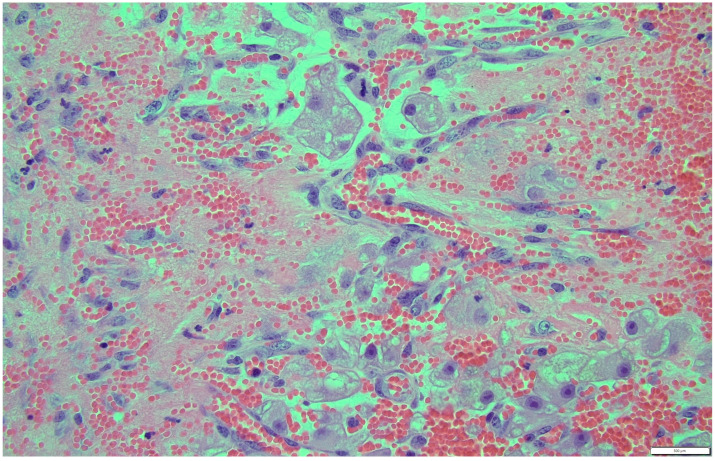
Mitotic rate was moderate in thecoma tissues—10 normal mitoses per 10 fields at 400× magnification (thecoma part), HE.

**Figure 14 animals-14-01307-f014:**
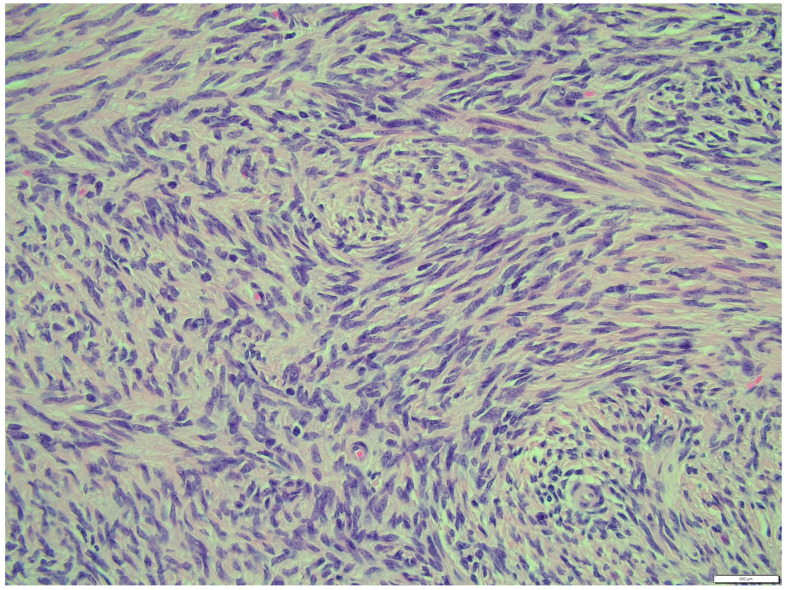
Spindle cell anisocytosis and anisokaryosis were mild-to-moderate, fibroma part. HE, 400× magnification.

**Figure 15 animals-14-01307-f015:**
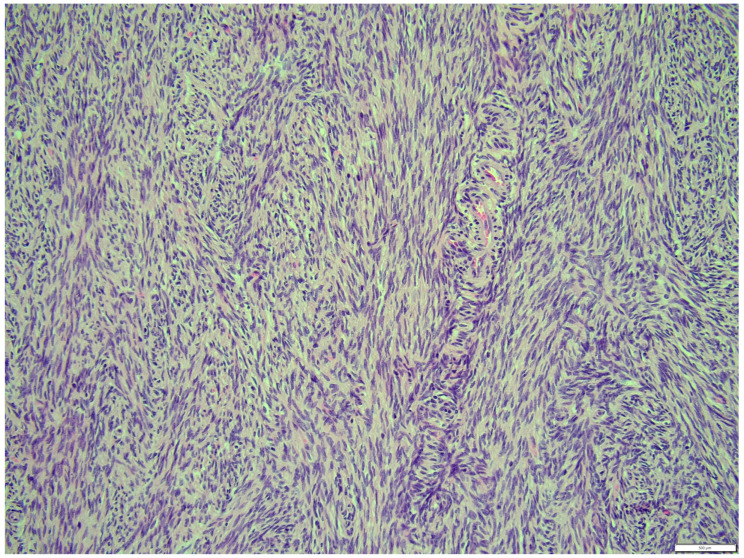
The spindle like cells were arranged in sweeping fascicles that are angled in a chevron-like or herringbone pattern, fibroma part. HE, 200× magnification.

**Figure 16 animals-14-01307-f016:**
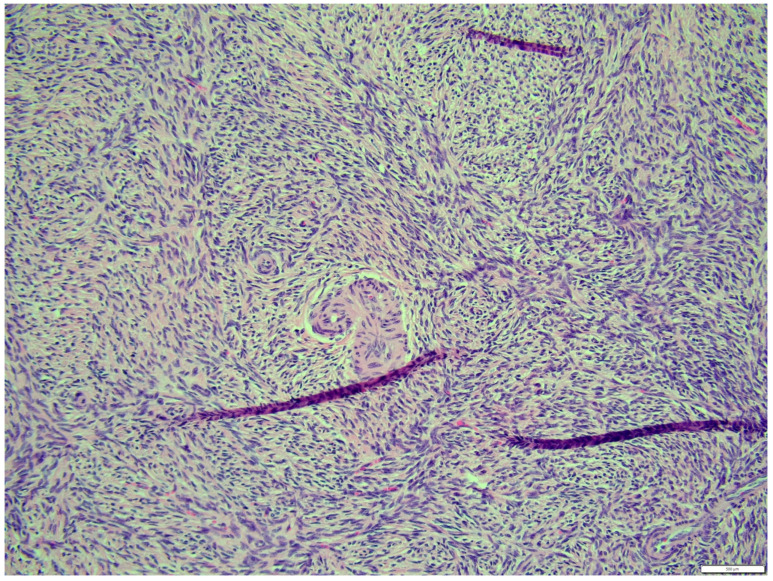
Storiform areas, fibroma part. HE, 200× magnification.

**Figure 17 animals-14-01307-f017:**
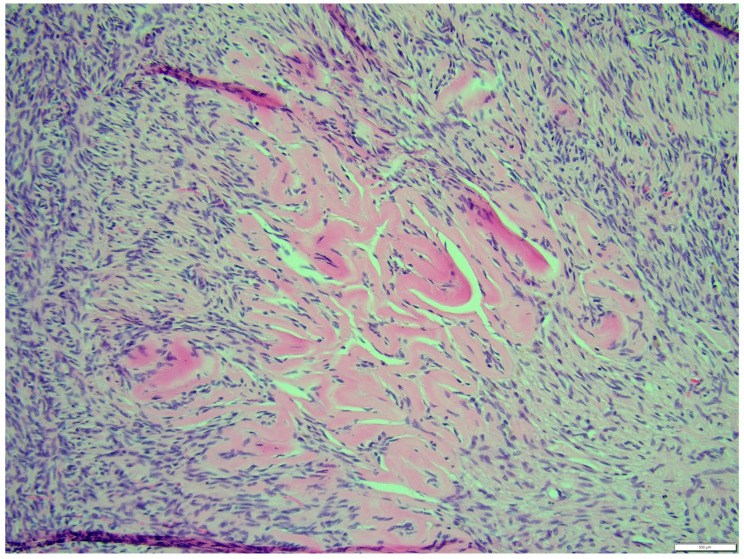
The stroma has delicate intercellular network of collagen, but a few areas show “keloid-like” sclerosis/hyalinization, fibroma part. HE, 200× magnification.

## Data Availability

The original contributions presented in the study are included in the article, further inquiries can be directed to the corresponding authors.

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
