# Peer review of "Ovarian Fibrothecoma in a Mare—Case Report"

_animals, 2024, doi:10.3390/ani14091307_

Round 1
Reviewer 1 Report
Comments and Suggestions for Authors
Dear authors,
The case described is interesting and the case under discussion is intriguing and addresses a pathology rarely described. However, I believe that some aspects need to be better elucidated; there is often a lack of rationale regarding the results and differences observed in your case compared to those described in the literature, both in veterinary and human fields. English editing is required.
Abstract: remove references in the abstract
- Line 27: Change not common with “uncommon”
- Line 27: change are divided into “are classified”
- Line 32: change sex cords stromal into sex cord stromal tumors
- Line 32: change into “Neoplasms like fibrothecoma present with limited morphological, clinical, ultrasonographic, and endocrine profile characteristics.”
- Line 37: rephrase and change into “After unilateral ovariectomy, the altered right ovary was diagnosed as fibrothecoma based on histopathological features”
Introduction
- Line 43: replace scarce with “limited”
Case history
- Line 57: please rephrase: According to the anamnesis, the mare experienced colic a month ago, which was resolved solely through a conservative approach
- Line 59: please provide more details about “all parameters”. Which parameters do you evaluate?
- Line 72:please rephrase into “On consecutive days, intravenous fluid and pain relief therapy were administered, along with fluids through the nasogastric tube, as the mare refused to drink.”. Please state the amount of fluids and the frequency of the administration
- Line 105: you stated that the ovarian mass was excited, I think it is not mandatory to specify that the specimens was obtained during surgery. You can say that the excided mass underwent to histopathologic examination.
- Line 112-113: replace applied with “administered”
- Line 115: where is the oedema? The right ovary was removed as you stated on line 102.
Diagnosis
- Line 123: I don’t understand what you mean by the following sentence: double histological image. Please clarify.
Discussion
- Line 158-162: please rephrase, the sentence is clear but a slight revision for smoother readability is requested.
- Line 171: the verb is missing
- Line 175: the verb is missing
- Line 179: what do you mean with “convenience and financial limitations? You speak about CT, but as you know, a CT scan of an adult horse abdomen is not possible
- Line 186: do you think that the mass you described could be appreciable with US? Why US was not performed? It is not clear why do describe in details the US appearance of the mass
- Line 198: as above. You talk about clinical signs in human, that are different from those described in your case. You have to argue
Comments on the Quality of English Language
English language editing is mandatory, there are also some grammar mistakes.
Author Response
Good day,
Thank you for your comments according to article “Ovarian Fibrothecoma in a Mare-Case Report”. I am sending you a corrected version of the article.
English language editing was made through recommended MDPI authors service. References was removed from “abstract” section.
Note: due to text editing, the numbers of the lines mentioned in the notes have changed.
According to another's editor comments, "diagnosis" part was significantly improved. Additional histological images, ultrasonographoc images at "case history" part were added.
- Line 24: common was changed with “uncommon”
- Line 24 “divided” was changed into “are classified”
- Line 27: “sex cords stromal” was changed into “sex cord stromal tumors”
- Line 28: was changed into “Neoplasms like fibrothecoma present with limited morphological, clinical, ultrasonographic, and endocrine profile characteristics.”
- Line 31: Changed into “After unilateral ovariectomy, the altered right ovary was diagnosed as fibrothecoma based on histopathological features”
Introduction
- Line 37: “scarce” was replaced into “limited”
Case history
- Line 50: was rephrased into” According to the history, the mare experienced colic a month ago, which was resolved solely through a conservative approach”
- Line 52: phrase all parameters were detalised
- Line 64: was rephrased into “On consecutive days, intravenous fluid and pain relief therapy were administered, along with fluids through the nasogastric tube, as the mare refused to drink.”. Amount of fluids and the frequency of the administration was added.
- Line 91: “ovarian mass was excited” was rephrased to “excided mass underwent to histopathologic examination”.
- Line 95-96: “applied” was replaced into “administered”
- Line 100: Mistake was corrected. Uterine, not ovarian edema was observed.
Diagnosis
- Line 116-117: “double histological image” was clarified to “Both macroscopically and histologically, two clearly distinct tumor zones were visible”
Discussion
- Line 185-186: Sentence was rephrased
- Phrase “convenience and financial limitations?” was removed. Information about CT was also removed.
- Line 186: The mass I described may be possibly appreciable with US. Ultrasound examination. In this case, ultrasonographic examination of reproductive system before laparotomy was not performed. Mare did not have history of estrous disorders and did not show discomfort during rectal palpation, so ultrasonographic examination was not indicated at this case.
Ultrasonography of ovarian mas may be beneficial and can provide additional information about neoplasm. On the other hand, ultrasonography can’t be used as a single diagnostic tool in diagnosing fibrothecoma, only in conjunction with histopathology, immunochemistry and/or hormonal level testing. Part about detailed US appearance was removed in “discussion” section was removed.
- Line 198: Clinical signs in human, that are different from those described in our case was removed.
Reviewer 2 Report
Comments and Suggestions for Authors
The assigned manuscript describes a case of biphasic tumor in a mare.
The case is interesting but the manuscript needs improvement before being considered for publication.
1) The first point concerns the numerous issues related to the English language, some verbs are missing. prepositions are sometimes not appropriate etc, so the manuscript needs professional editing or, at least, revision by a native speaker. This especially will determine my judgment of "major revision"
2) The second important point concerns the tumor itself which, histologically, needs to be shown even at a lower magnification because the histological photo in which the two components are shown, while well done does not let one understand the organization of the tumor. So insert a low magnification photo. It doesn't matter if it won't be wonderful, however, it will give the idea of the biphasicity of the tumor.
3) The third point concerns the histological description of the tumor which is quite complete but not conformed according to the descriptive steps of the European ( or American) College of Veterinary Pathologists. According with these steps, you produce successive sentences, always the same in their order, and first you decribe the presence of a tumor in the organ and how the tumor fit into the organ. In this case the neoplasm effaced and replaced the entire ovary. Next the tumor neoformation is described, which, in this case, appears to consist of 2 c omponents (thecal and fibromatous). Then you go on to describe the individual components, first one and then the other, such as "thecal component was composed by neoplastic cells.... arranged in .... (architecture-even considering that they sometimes circumscribe blood-filled lacunae) and then move on to describing the neoplastic cells by indicating: cell shape, size, borders (distinct or indistinct or variable), then describing the nucleus/cytoplasm ratio ( in favor of the cytoplasm? Of the nucleus? Intermediate?) and the amount of cytoplasm (moderate in this case), the possible presence of vacuoles and dye affinities (faintly or moderately or strongly exonophilic?). Then you turn to the nucleus by describing its shape or range (e.g., round to oval) and then decribe its appearance, (with condensed or dispersed chromatin?) and to the presence or absence of a prominent nucleolus. Then you enter whether there is anisocytosis and/or anisokaryosis and their degree (mild? Moderate? No?) and then describe the mitoses (atypical? Maybe not), their range in 10 fields at 400X and, finally the mitotic count = sum of mitoses in the 10 fields considered. Better still would be to count them at 400X in 2.37 mm2. Lastly, various other features such as inflammatory infiltrates, necrosis, hemorrhage, of course if present, are entered.
4) Fourth point, discussion: this is very long because it includes comments on the whole colic part (from line 232). This part is not relevant to the case and the literature on colic is very abundant and the treatment very well known, so it should be removed in my opinion.
Other comments
- Lines 5-12: no need to multiply affiliation if all authors belong to the same institution
- Line 29: germ cells not germinal
- Line 33: poor morphology? What is meant by this?
Comments on the Quality of English LanguageAlready included in general comments. English needs editing, moderate but thorough because sometimes verbs are missing or incorrect in person and prepositions are not always appropriate
Author Response
Good day,
Thank you for your comments according to article “Ovarian Fibrothecoma in a Mare-Case Report”. I am sending you a corrected version of the article. English language editing was made through recommended MDPI authors service.
Note: due to text editing, the numbers of the lines mentioned in the notes have changed.
“Diagnosis part” was significantly improved with histological pictures at lower magnification.
In order to improve "case history" part, additional ultrasonographic images of reproductive system ultrasound post unilateral ovariectomy was added.
Histological description of the tumor was improved according to the descriptive steps of the European ( or American) College of Veterinary Pathologists- “diagnosis” part.
Colic part at the “ discussion” was removed.
- Lines 5-12: redundant affiliations was removed
- Line 25: “germ cells” was changed into ”germinal”
- Line 27-28: “poor morphology” was changed to “Neoplasms like fibrothecoma present with limited morphological, clinical, ultrasonographic, and endocrine profile characteristics.” According to another editor comments.